META-RESEARCH

# Author-level data confirm the widening gender gap in publishing rates during COVID-19

**Abstract**  Publications are essential for a successful academic career, and there is evidence that the COVID-19 pandemic has amplified existing gender disparities in the publishing process. We used longitudinal publication data on 431,207 authors in four disciplines - basic medicine, biology, chemistry and clinical medicine - to quantify the differential impact of COVID-19 on the annual publishing rates of men and women. In a difference-in-differences analysis, we estimated that the average gender difference in publication productivity increased from –0.26 in 2019 to –0.35 in 2020; this corresponds to the output of women being 17% lower than the output of men in 2109, and 24% lower in 2020. An age-group comparison showed a widening gender gap for both early-career and mid-career scientists. The increasing gender gap was most pronounced among highly productive authors and in biology and clinical medicine. Our study demonstrates the importance of reinforcing institutional commitments to diversity through policies that support the inclusion and retention of women in research.

**EMIL BARGMANN MADSEN, MATHIAS WULLUM NIELSEN, JOSEFINE BJØRNHOLM, RESHMA JAGSI AND JENS PETER ANDERSEN***

**\*For correspondence:**
jpa@ps.au.dk

**Competing interest:** The authors declare that no competing interests exist.

## Introduction

Gender disparities in academic publishing have widened during the COVID-19 pandemic. The proportion of preprints and manuscript submissions with women as authors has decreased (*Cui et al., 2021*; *Kibbe, 2020*; *Mogensen et al., 2021*; *Squazzoni et al., 2020*; *Williams et al., 2021*), as has the proportions of preprints and published articles with women as either the first author or the senior author (*Andersen et al., 2020*; *Inno et al., 2020*; *Lerchenmüller et al., 2021*; *Muric et al., 2021*; *Ribarovska et al., 2021*). Gender gaps in self-reported research activities have also increased (*Andersen et al., 2020*; *Inno et al., 2020*; *Lerchenmüller et al., 2021*; *Muric et al., 2021*; *Ribarovska et al., 2021*). However, the longitudinal effects of the pandemic on differences in annual publication outputs remain uncertain. In this study, we used individual-level panel data on the publication activities of 431,207 authors globally to

quantify the differential impact of COVID-19 on the publishing rates of women and men.

Research on gender and publication productivity suggests that women (on average) publish fewer articles than men (*Mairesse and Pezzoni, 2015*), although the magnitude of this difference varies by career stage, discipline and country, and has diminished over time (*Huang et al., 2020*; *Sax et al., 2002*; *Xie and Schauman, 2005*). The gender imbalance in publishing rates should be understood in the context of broader disparities in the science system. Structural variables such as employment rank, access to resources, university prestige, appointment type, teaching loads (*Eagly, 2020*; *Taylor et al., 2006*) and available time for research (*Guarino and Borden, 2017*; *Leišytė, 2016*) all partially explain the observed gender imbalances in publication productivity (*Allison and Long, 1990*; *Bland et al., 2006*; *Xie and Shauman, 1998*). In addition, research finds that women scientists (compared to men) tend to span more topics in their research activities, face stricter editorial standards in peer reviewing

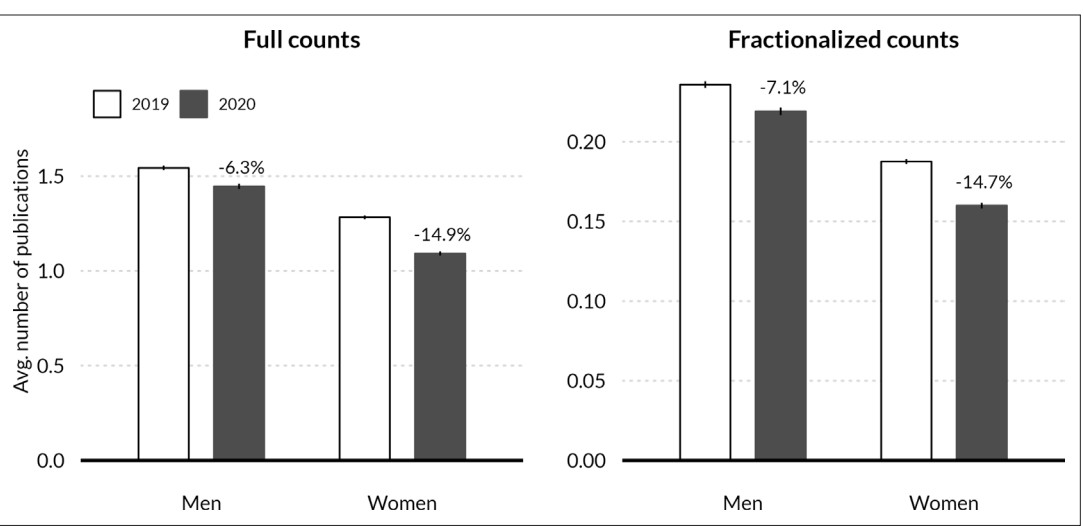

**Figure 1.** Average publication output by gender and year. Differences are in percentages of average publication rates in 2019. Results are presented for full and fractionalized publication counts. Men experience a smaller productivity decrease in 2020 compared to 2019 (6.3%) than women (14.9%) using full counts of publications. For fractional counts (each paper counts as a fraction of the number of co-authors), the difference in decrease is greater, with a 7.1% decrease for men and 14.7% decrease for women. Average publication counts are presented with 99% confidence bounds.

(*Hengel, 2017*), and take on greater shares of parenthood responsibilities (*Derrick et al., 2021*), which also likely perpetuate publishing disparities.

Recent research has identified two primary mechanisms through which the pandemic may have amplified existing disparities in publishing (*King and Frederickson, 2021*). First, evidence from national and international surveys indicates that women scientists have taken the lion's share of the extra childcare and domestic responsibilities imposed by lockdowns of schools and daycares (*Deryugina et al., 2021*; *Staniscuaski et al., 2021*; *Yildirim and Eslen-Ziya, 2020*). According to surveys of self-reported research activities, women scientists – especially those with young dependents – have seen notable productivity decreases in the wake of the pandemic (*Deryugina et al., 2021*; *Myers et al., 2020*; *Staniscuaski et al., 2021*). Second, transitions to online teaching during university lockdowns required extra hours of planning and preparation and may have affected women scientists more than men due to observed disparities in average teaching loads (*Barber et al., 2021*; *Eagly, 2020*; *King and Frederickson, 2021*; *Taylor et al., 2006*). Survey-based evidence from the United States also indicates that the extra time spent on teaching partially accounts for observed decreases in scientists' self-reported publication rates (*Barber et al., 2021*). In clinical medicine, service demands related to care for COVID-19 patients and transitions to virtual care delivery for many others may also have disproportionately affected women, who are more likely to be represented on clinician-educator rather than traditional tenure tracks at medical schools (*Mayer et al., 2014*).

This study is, to our knowledge, the first to quantify the differential impact of COVID-19 on the annual publishing rates of women and men. We used a linked dataset of 431,207 authors and 2,113,108 publications and a difference-in-differences specification to estimate how the gender difference in average publishing rates changed from 2019–2020.

We rely on author-disambiguated publication data from Clarivate's Web of Science, restricting our focus to scientists with >2 publications within basic medicine, biology, chemistry and clinical medicine. We chose these fields as they are well-represented in Web of Science (more than 90% of references are included in Web of Science), their primary knowledge production mode is through journal publication (unlike, for example, computer science, many fields of engineering, and the humanities), research is comparatively collaborative (although some areas of clinical research have somewhat more authors), publishing is relatively fast (compared to, for example, the social sciences). Basic medicine, biology and clinical medicine also have some of the highest shares of women scientists in the natural sciences.

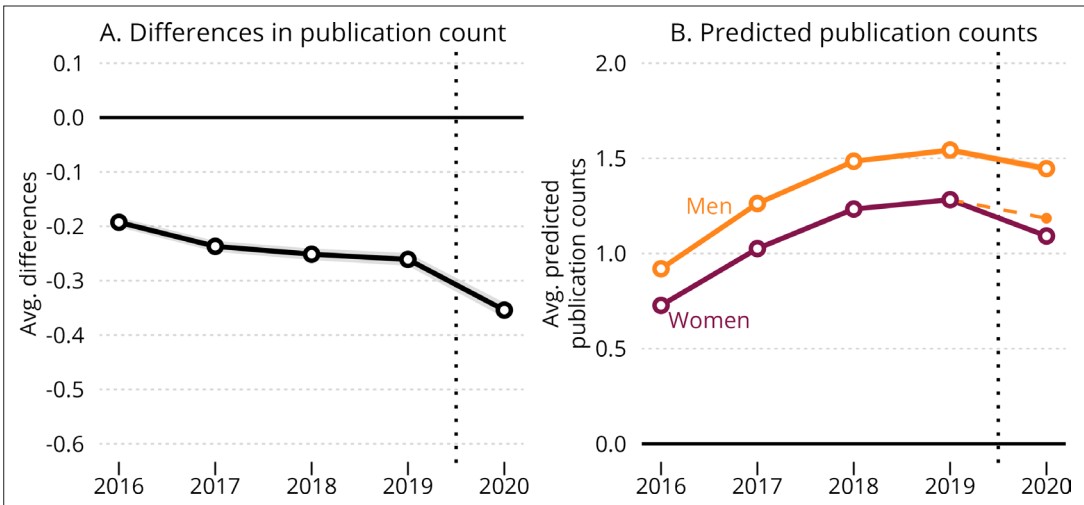

**Figure 2.** Dynamic effects of the COVID-19 pandemic on women's and men's publication productivity. Panel A shows the estimated average gender difference in publication rates by year. Each point shows the relative difference between men and women per year, with 99.9% confidence bounds shown as a gray area around the line. From 2019 to 2020, the average-marginal gender difference increased from –0.260 (17% lower output for women) to –0.354 (24% lower output for women). Panel B shows the predicted publishing rates for men and women authors, with solid lines showing the trend per gender, and the dashed, orange line showing the counterfactual trend for women if they had similar 2019–2020 trajectories as men (i.e. the trend for men is projected to the 2019 estimate for women). The difference between the dashed line and the straight line in Panel B specifies the average treatment effect for women. Point estimates are reported with 99.9% confidence bounds, with robust standard errors clustered at the individual-author level. For information on how average marginal and predicted values are calculated, please refer to Materials and Methods: Difference-in-Differences model.

The online version of this article includes the following source data and figure supplement(s) for figure 2:

**Source data 1.** OLS linear regression with full count as dependent variable.

**Source data 2.** OLS linear regression with fractional count as dependent variable.

**Source data 3.** Poisson regression with full count as dependent variable.

**Source data 4.** Negative binomial regression with full count as dependent variable.

**Figure supplement 1.** Corresponding analysis with fractional counts.

---

We report annual, per-author publishing rates based on a full and fractional counting. The full counting gives the raw sum of all papers published by a scientist in a given year. The fractional counting gives the sum of the reciprocal of the number of authors per paper published by a scientist.

## Results

The following results use a main sample consisting of two scientist cohorts, one with first publication year in 2009 or 2010 ("mid-career", n = 137,767) and one with first publication year in 2016 or 2017 ("early-career", n = 293,440). Unless mentioned otherwise, the combined cohort (n = 431,207) is used. A third, counterfactual cohort (n = 276,793) is used to contrast the early-career sample, as a means of estimating the expected attrition in the early-career stage, when a proportion of scientists leave academia. Each analysis referring to a

"treatment", indicated in figures as a dotted line between 2019 and 2020, refers to the changes in working environments in 2020 due to the COVID-19 pandemic.

### Descriptive results

Our analysis suggests that gender disparities in annual publication outputs have widened during COVID-19. A descriptive comparison of changes in publishing rates in 2020 compared to 2019 (*Figure 1*) indicates a 15% decrease in women's average full- and fractional-count publication output and a 6%–7% decrease in men's average full- and fractional-count publication output.

### Difference-in-differences estimates

*Figure 2* displays the dynamic effects of the COVID-19 pandemic and summarizes the main result of the difference-in-differences estimation. As shown in panel A, the gender difference in

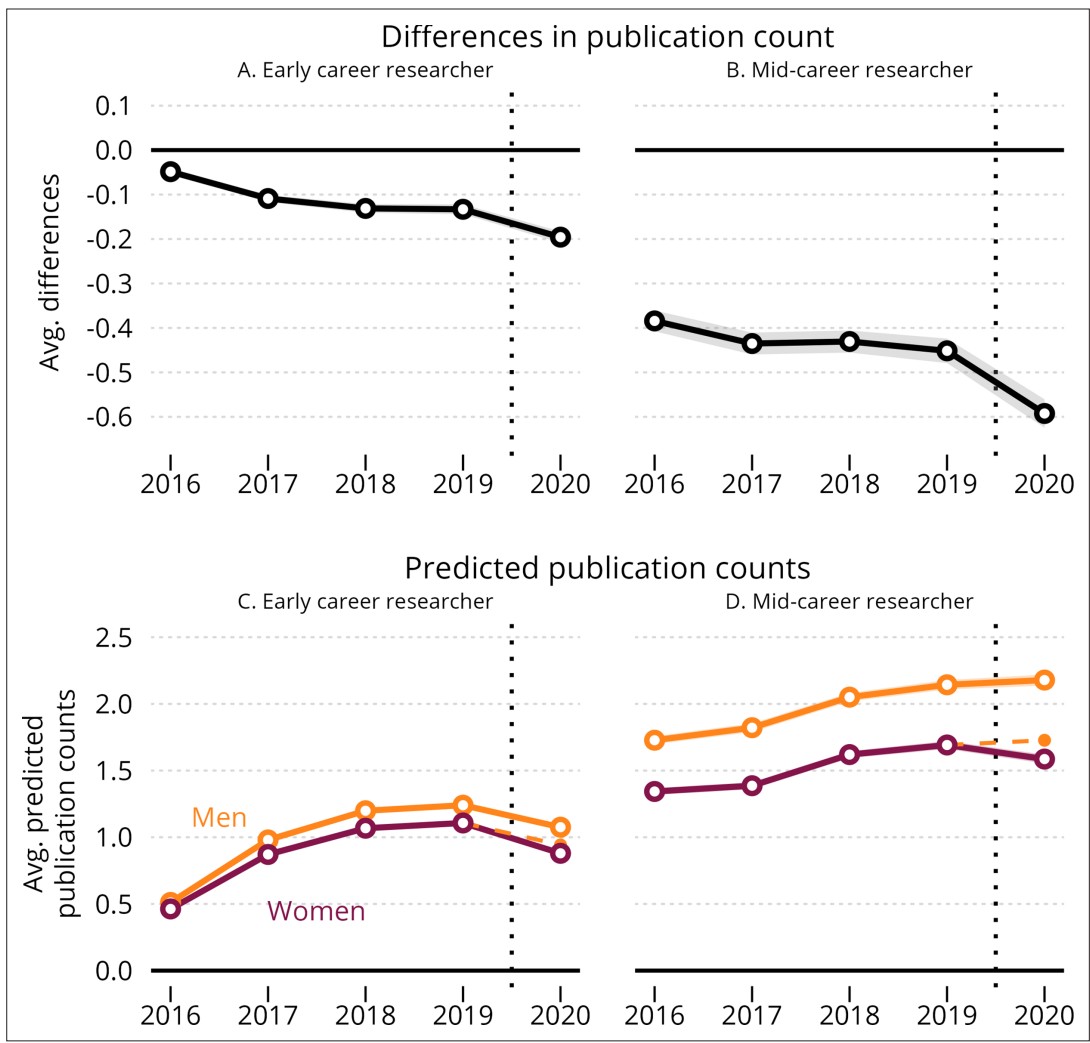

**Figure 3.** Dynamic effects of the COVID-19 pandemic on the average gender gap in annual publishing rates, by career age. Panels A and B show the estimated average gender difference in full-count publication rates by year for early-career and mid-career researchers. Panels C and D show men's and women's predicted full-count publication rates per year by author status (early-career vs. mid-career researcher). Point estimates are reported with 99.9% confidence bounds and robust standard errors clustered at the individual-author level. For information on how average marginal and predicted values are calculated, please refer to Materials and Methods: Difference-in-Differences model.

The online version of this article includes the following source data and figure supplement(s) for figure 3:

**Figure supplement 1.** Corresponding analysis with fractional counts.

**Figure supplement 2.** Corresponding analysis with counterfactual sample.

**Source data 1.** OLS linear regression of the early-career sample, with full count as dependent variable.

**Source data 2.** OLS linear regression of the mid-career sample, with full count as dependent variable.

**Source data 3.** OLS linear regression of the early-career sample, with fractional count as dependent variable.

**Source data 4.** OLS linear regression of the mid-career sample, with fractional count as dependent variable.

annual publishing rates remained relatively stable between 2017 and 2019 (implying parallel trends prior to COVID-19), while increasing in 2020. From 2019 to 2020, the average-marginal gender difference increased from –0.260 (corresponding to a 17% lower output for women than for men) to –0.354 (corresponding to a 24% lower output for women than for men) in full-count output. *Figure 2—figure supplement 1* presents results from a complementary analysis with fractional-count publication output as outcome and shows a change in the average-marginal gender

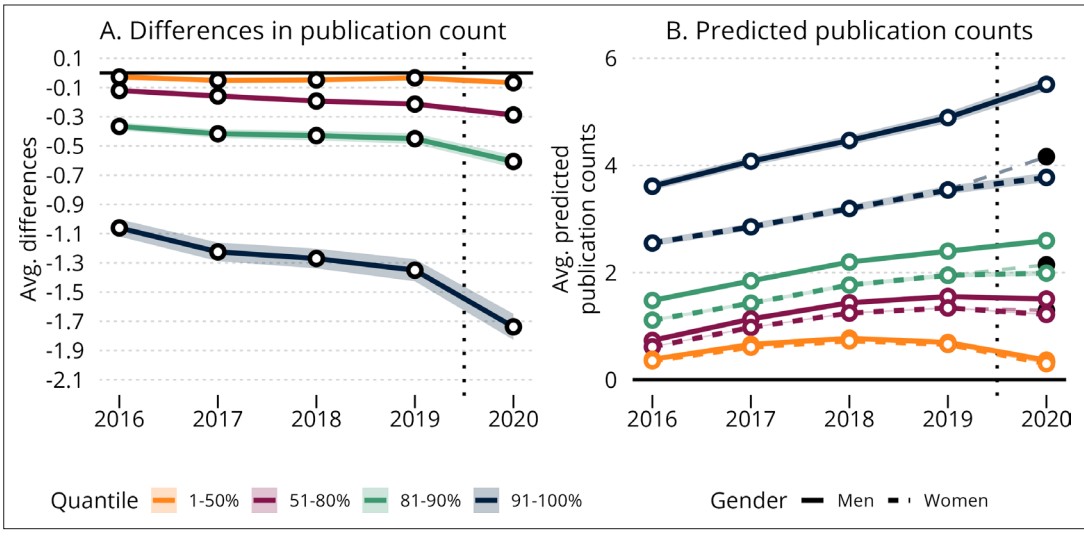

**Figure 4.** Stratified effects of the COVID-19 pandemic on the average gender gap in annual publishing rates. Panel **A** shows the estimated average gender difference in publication rates by year. Panel **B** shows the predicted publishing rates for men and women authors. In each panel, scientists are divided into strata according to their total number of publications in the period 2016–2020. The difference between the thinner, dashed line with the black circle in 2020 and the thicker, dashed line with hollow circles in panel B specifies the average treatment effect for women. Point estimates are reported with 99.9% confidence bounds and robust standard errors clustered at the individual-author level. For information on how average marginal and predicted values are calculated, please refer to Materials and methods: Difference-in-differences model.

difference from –0.048 (corresponding to a 22% lower output for women than for men) to 0.059 (corresponding to a 27% lower output for women than for men).

To verify that the change in the gender productivity gap was in fact due to COVID-19 and did not represent a more generic dip in women's productivity (compared to men's) during the fifth year of their publication career, we ran a counterfactual analysis for a sample of researchers, who published their first paper in 2011. For this sample, we observed a small but consistent annual increase in the marginal gender difference across years (from 2011–2015). In this case, the gender difference in productivity increased by 1/20 of a full publication (full count: –0.05, 99% CI: –0.0665; –0.0337) between year four (2014) and five (2015), amounting to 53% of the treatment effect observed in *Figure 2*.

### Career-stage differences

Research suggests that the working conditions of early-career women scientists have been especially affected by the pandemic (*Andersen et al., 2020*; *Krukowski et al., 2021*). We examined this question by conducting sub-group analyses by career-age. As shown in *Figure 3* the widening gender gap was salient for early-career scientists with four years of publication experience as

well as for mid-career scientists with ten years of publication experience. From 2019 to 2020, the average marginal publication disadvantage for early-career women increased from –0.133 (corresponding to an 11% lower output for women than for men) to –0.20 (corresponding to an 18% lower output for women than for men) in full-count output. In comparison, the average marginal publication disadvantage for mid-career women changed from –0.452 (corresponding to a 21% lower output for women than for men) to –0.592 (corresponding to a 27% lower output for women than for men). This is a relative increase in the gender gap of 61% for early-career scientists and 29% for mid-career scientists. We obtained comparable results in an age-differentiated analysis with fractional-count publications as outcome (*Figure 3—figure supplement 1*).

### Productivity-dependant differences

As indicated in *Figure 4* panel A, the effect of the pandemic on women's and men's publishing rates also varied considerably across different strata of the publication-productivity distribution. Indeed, a considerable share of the average marginal gender difference appeared to be attributable to differences occurring among the top-10% most prolific men and women authors. In contrast, changes in the average gender gap were marginal

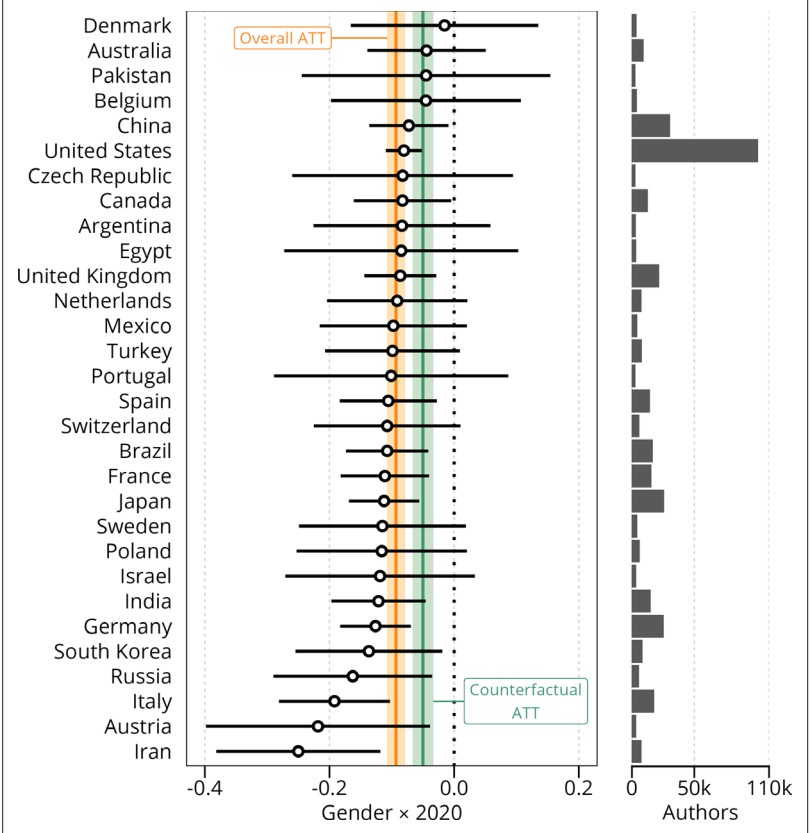

**Figure 5.** Gender differences in full publication productivity by country, 2019 vs 2020. The hollow circles show the gender differences per country in full publications counts in 2020 relative to 2019, with error bars showing the 99% confidence intervals based on robust clustered standard errors. Countries are ranked by the estimated gender difference. The horizontal histogram shows the distribution of authors from each country, showing that the vast majority are from the USA. We only list the first 30 countries by number of authors, comprising 90% of authors in our sample. The orange and green lines and bands show the overall treatment effect on the sample and the counterfactual sample. (ATT is the Average Treatment effect on the Treated).

The online version of this article includes the following source data and figure supplement(s) for figure 5:

**Figure supplement 1.** Lockdown severity, summed indicators.

**Figure supplement 2.** Lockdown severity, maximum indicators.

**Source data 1.** OLS linear regression of counterfactual sample, with full count as dependent variable.

**Source data 2.** OLS linear regression of counterfactual sample, with fractional count as dependent variable.

**Source data 3.** Coefficients and standard errors relative to 2019 for the 30 countries with most authors in the dataset.

for authors below the 80th percentile of the publication distribution. This can clearly be seen in panel B, where the trends for men per quantile in 2019–2020 (solid, black dots) is projected unto the same trends for women (hollow dots). While the differences in trends below the 80th percentile are not visible in the figure, and the absolute differences are very small, the relative

differences are noticeable. At the highest decile, the average difference increases from −1.35 (corresponding to 23% lower output for women) to −1.74 (31% lower output for women) from 2019–2020,, which is a relative change of 22.3%. Correspondingly the relative change is 25.8% in the 81st to 90th percentile and 25.9% in the 51st to 80th percentile.

### Country-level differences

The estimated change in the magnitude of the gender gap also varied across countries (*Figure 5*), with the smallest changes observed in Denmark, Australia, Pakistan and Belgium, and the largest increases found in Russia, Italy, Austria and Iran. The horizontal bar diagram to the right in *Figure 5* shows that the vast majority of scientists are from the USA. This means that the average treatment effect on the treated ($ATT$) also gravitates towards the effect observed for the US population. Surprisingly, the estimated effects at the country-level were only weakly and inconsistently correlated with the severity of COVID-19 restrictions (*Figure 5—figure supplement 1* and *Figure 5—figure supplement 2*).

### Discipline-level differences

As a final step in the analysis, we disaggregated results by discipline. As shown in *Figure 6* panel A, the widening gender gap was persistent across all four disciplines but with markedly larger effects observed for clinical medicine (Average marginal gender difference = −0.117, CI: –0.138––0.095) and biology (Average marginal gender difference = −0.089, CI: –0.117––0.063) compared to basic medicine (Average marginal gender difference = 0.058, CI: –0.093––0.022) and chemistry (Average marginal gender difference = 0.062, CI: –0.100––0.023). *Figure 6* panel B specifies the representation of authors according to their position in the publication-productivity distribution, across the four disciplines. As shown in the figure, we observe an over-representation of highly productive authors in clinical medicine implying that the large average marginal gender difference effect observed for this discipline may partially be driven by a higher proportion of prolific scientists.

### Robustness checks

We conducted two (*Figure 7*) placebo tests, simulating a placebo pandemic incident between 2017–2018 and 2018–2019. shows the difference-in-differences estimate for both full and fractionalized publication counts. In both cases, the

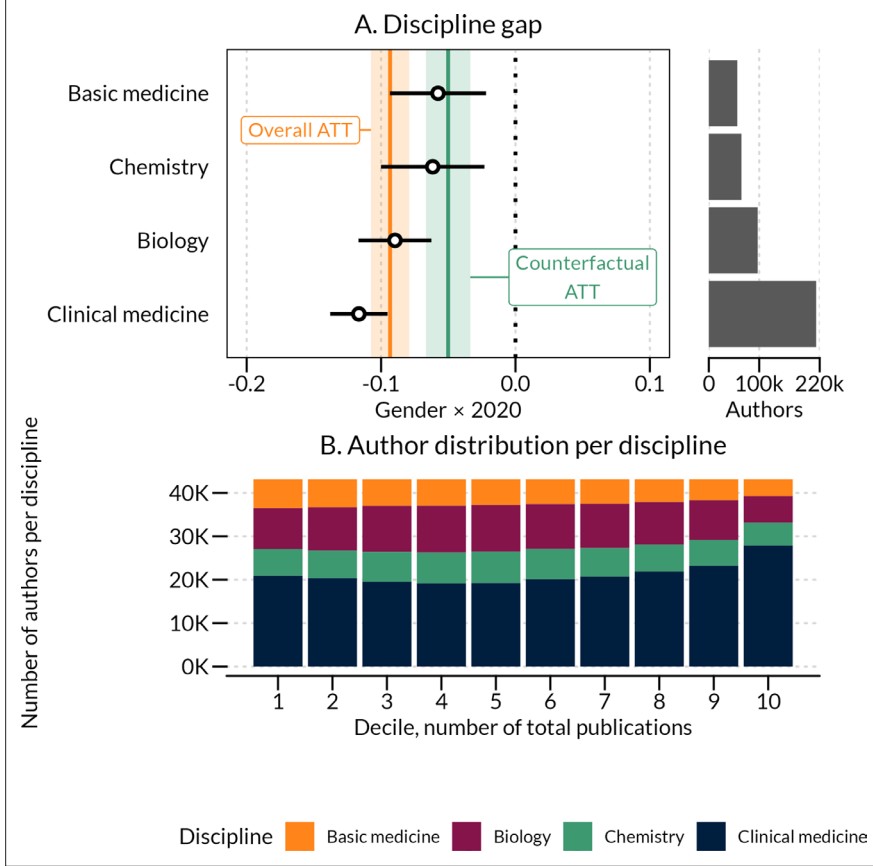

A. Discipline gap

B. Author distribution per discipline

**Figure 6.** 2020 gender differences in full publications counts relative to 2019, across the four disciplines comprising in our sample. Difference-in-differences estimate from *Figure 6—source data 1*. 99% confidence intervals based on clustered standard errors are shown. Histograms show the distribution of authors who mainly publish within a given discipline, and orange and green lines and bands show the overall treatment effect on the sample and the counterfactual sample from *Figure 2—source data 1* and *Figure 5—source data 1*. Panel **B** shows the distribution of authors per discipline in deciles of total publications over the time period.

The online version of this article includes the following source data for figure 6:

**Source data 1.** Coefficients and standard errors relative to 2019 for the four disciplines.

estimates are very small in magnitude (ranging from 7%–17% of our 2020 estimate, $\delta_{t=0}$), and only statistically significant for the 2017–2018, full count, estimate at the 99% level (the 2017–2018 estimate is significant for the fractionalized count at a 95% level). Taken together, there does not appear to be a substantial difference in publication counts in the immediate years prior to the onset of the pandemic.

We also check whether there are changes in the position in the author byline of women authors (see *Figure 7B*). We first observe, that the share of women first authors is higher than expected, considering the share of women in total. Some variation occurs over time, but there are no changes from 2019–2020 which could

indicate a general shift in women appearing less often as first authors than before the pandemic.

## Discussion

In this paper, we estimated the differential impact of COVID-19 on the annual publication rates of women and men in 2020 compared to 2019. Using individual-level panel data on a global sample of 431,207 authors, we observed small but consistent average increases in the gap between women's and men's annual publishing rates. This finding is consistent with extant research suggesting amplified gender disparities in manuscript submissions, first and last author-ships, and self-reported research activities during COVID-19. However, unlike prior studies, we find that the gendered effects of COVID-19 are salient for early-career-scientists with four years of publication experience as well as for mid-career scientists with ten years of publication experience. While the numerical increase in the gender gap is largest for mid-career scientists, the relative change in the gender gap is biggest for early-career scientists. Moreover, we add to existing evidence by showing that the increase in the gender gap (in absolute terms) was most pronounced among highly productive authors and scientists working in clinical medicine and biology. Lastly, the widening gender gap appears to represent a genuine decline in publication productivity and not just a shift in author roles, as women continue to first author publications at similar rates as in prior years (*Figure 7*).

Despite clear country variations in the observed effects, we found negligible and inconsistent associations between local COVID-19 restrictions and estimated changes in the productivity gender gap. Further, the ordering of countries in *Figure 5* does not seem to suggest that the gender-differentiated changes in productivity rates vary systematically according to a country's level of gender equality, welfare model, or infection rate.

Taken together, these results indicate that the publication productivity of already prolific women scientists have been affected the most by the pandemic. Those designing interventions to promote equity in academic science and medicine should strive to understand the reasons why highly prolific men appeared able to maintain their annual publication rates while highly prolific women were not. Prior research suggests that it is possible that men with the highest levels of productivity may have been more likely to have been rewarded with access to

**Table 1.** Seven indicators of COVID-19 lockdown severity.

|  | Sum indicator | Count of maximum values |
| --- | --- | --- |
| School lockdowns | + | + |
| Workplace lockdowns | + | + |
| Stay at home requirements | + | + |
| Stringency index | + | - |

additional workplace supports, such as endowed professorships, in recognition of their achievements (*Gold et al., 2020*). If so, this might have served as a cushion against the impact of the pandemic on those individuals. Moreover, if institutions prioritized protecting a few "superstar" researchers from teaching or clinical demands without clear processes for identifying which individuals received preferential treatment, the vast literature on unconscious bias suggests that such efforts might preferentially have protected outstanding men as compared to similarly outstanding women (*NASEM, 2007*). Prior research also suggests that high-achieving women scientists may be more likely than their male peers to state that their partners' careers take priority (*Mody et al., 2022*). Indeed, it is possible that high-achieving men scientists' partners may be particularly likely to be willing to make sacrifices in their own careers to take on additional domestic labor to allow continuation of their extraordinary partners' work. If partners of extraordinarily productive women scientists are less willing to do so, and if this difference is even more marked than any differences that may exist when a scientist is less highly productive, this could also serve as a mechanism to drive the differences observed. Further research is necessary to investigate these and other possibilities.

The amplified effect in clinical medicine may be due to the dual research and clinical roles taken on by scientists in this discipline. Early research suggested that initial funding for COVID-19 related research was biased toward applications from men (*Witteman et al., 2021*), supporting a hypothesis that women spent disproportionally more time on clinical work or other demands around the time of the outbreak. However, further research is required to provide conclusive evidence on this question. The consequences of a systematically biased change in the work priorities for men and women in particularly clinical medicine can potentially reach far

beyond the individual careers of those women affected by it. Research suggests a positive association between women's participation as leading authors in medical research and a study's likelihood of including sex and gender as analytical variables (*Nielsen et al., 2017*). The omission of gender and sex analysis has been widespread in COVID-19-related clinical trials (*Brady et al., 2021*), despite early evidence of sex-differences in the prognosis and outcome of the disease.

The widening gender-gap in publishing may be a detectable symptom of larger setbacks on issues of gender equity in science (*King and Frederickson, 2021*). Indeed, recent research also shows widening gender disparities in research project initiation (*Gao et al., 2021*) and clinical-trial leadership (*Cevik et al., 2021*).

Our study demonstrates the importance of reinforcing institutional commitments to gender equity through policies that support the inclusion and retention of women researchers (*Andersen et al., 2020*; *Fulweiler et al., 2021*; *King and Frederickson, 2021*; *Narayana et al., 2020*). While our study focuses on gender, other marginalized groups are likely to suffer from similar setbacks, potentially to an even higher degree. These groups are generally under-studied in the the literature on productivity gaps, as they are much more difficult to identify quantitatively. Further research, with reliable data on especially ethnicity, and with an inter-sectional perspective is needed.

Data on individual publication rates gives us a better estimate of the effects of the pandemic on researcher productivity than most previously published analyses focusing on publication-level effects. Despite this, the data do not allow us to disentangle how much of the widening gender gap is due to attrition. If the relative share of women scientists opting out of an academic career is higher in 2020 compared to 2019, this may inflate the observed change in productivity. Future research should examine the potential changes in women's and men's attrition rates in closer detail. Further, the counter-factual analysis presented in *Figure 3—figure supplement 2* suggests a consistent increase in the size of gender productivity gap over time with a marginal annual change in the gender difference from year four to five amounting to 53% of the treatment effect observed in our main analysis. The estimated change from a 17% lower output for women than men in 2019 compared to 24% percent lower output for women than men in 2020 should thus be interpreted with some caution. However, both mechanisms - lower publication productivity and

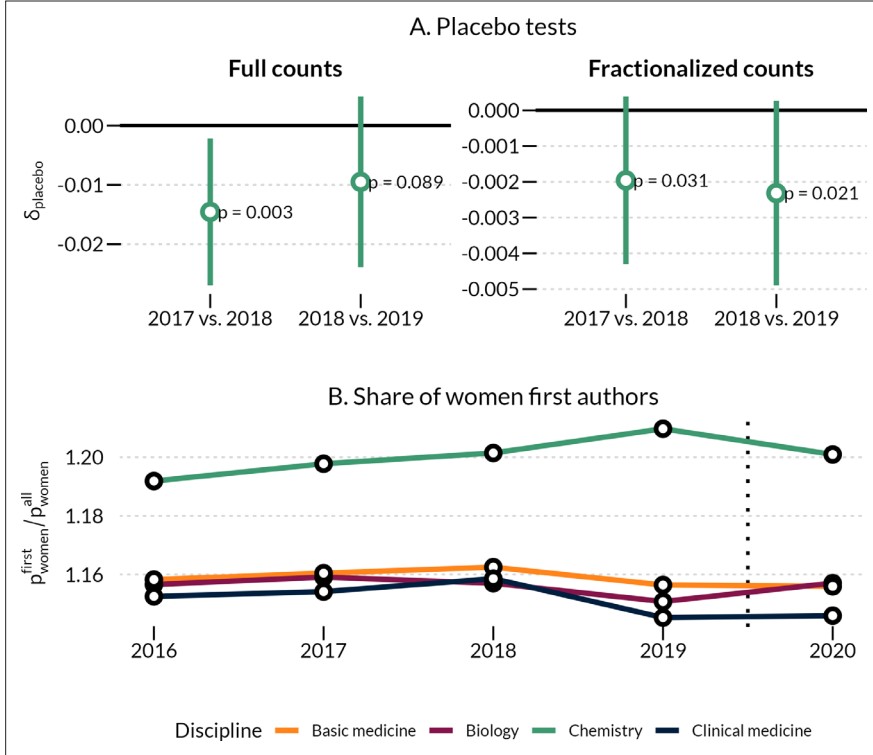

**Figure 7.** Test against hypothetical placebo pandemic in 2018/2019 (**A**) and changes in women as first authors (**B**). (**A**) Difference in differences of publication productivity for placebo tests. Points show the difference in publication productivity for women relative to men for two placebo periods, using both full publication counts and fractionalized counts. Estimates are based on *Figure 7—source data 1* and *Figure 7—source data 2*. Errorbars are 99% confidence intervals, with accompanying p-values based on clustered standard errors. (**B**) Ratio of women's first author share to women's share of all authorships. Each line shows the share of women who occupy the first author position divided by women's share of all authorships by year for each of the four disciplines. A ratio > 1 shows a greater share of women first authors relative to all women's authorships. Authorships counts are made for a larger sample than used in the main analysis, comprising all authorships registered in the Web of Science for each discipline and year.

The online version of this article includes the following source data for figure 7:

**Source data 1.** OLS linear regression of full and fractional count as dependent variable, placebo test of 2017 vs 2018.

**Source data 2.** OLS linear regression of full and fractional count as dependent variable, placebo test of 2018 vs 2019.

attrition - result in lower total publication outputs for women and lead to enlarged gender disparities. While we can not currently estimate the relationship between the two mechanisms, the conclusions above remain the same.

Our study design has four limitations. First, our analysis focused on annual publishing rates, which may obscure some of the potential effects of e.g. school closures on the immediate publishing rates. A more granular analysis of monthly publishing rates may reveal a more direct correlation between lockdowns and decreased publishing rates. However, information on when something is published is not available on a monthly basis for a large proportion of articles, and information on submission and review dates are even harder to obtain, often completely missing. Further, many of the delays occurring in the publishing process are out of the hand of authors and thus unrelated to the lockdown effect that they may be experiencing. By looking at annual data, we can estimate a more reliable effect overall. We strongly encourage publishers to make available transparent, open machine- and human-accessible information on which date a manuscript was received, reviewed, revised, accepted and published. Similarly, the weak relationship between country-level gender gaps and the severity of lockdown policies could be due to aggregation. Using survey data on self-reported time-use, *Deryugina et al., 2022* show that e.g. the fraction of days with at least partial primary school closures negatively affected time loss for women researchers relative to men in the period Feb. 16 - July 31, 2020. To compare our yearly publication data with lockdown severity, we aggregated day-to-day data on school closures, workplace closures, stay at home requirements, and overall lockdown severity across the entire year of 2020.

Second, the author-disambiguation approach used to establish individual-level panel data unavoidably introduces some level of uncertainty into our analysis, and errors are more likely to occur for individuals with East Asian names (*Nielsen and Andersen, 2021*) (see Materials and Methods). The country-specific evidence for China and South-Korea (*Figure 5*) should thus also be interpreted with caution.

Third, the gender-assignment algorithm used in this study did not infer the gender of 20% of the author sample. This introduces potential sampling bias into our analysis. Moreover, the algorithm reduces author gender to a binary category (woman or man), but not all individuals identify as women or men. Despite this clear limitation, we find the algorithm useful in quantifying COVID-19-related disparities on a large scale (*D'Ignazio and Klein, 2020*).

Fourth, academic publishing is a slow endeavor, and article submissions may undergo many rounds of revisions before they are published (*Homolak et al., 2020*). This introduces two types of potential bias into our analysis: (a) some of the articles published in 2020 are based on research conducted in 2019; and (b) some of the research conducted in 2020 will not appear in print before 2021, or later. Thus, in the coming years, scientists should continue

to monitor disparities in women's and men's publishing rates.

In science, even small negative kicks or setbacks may add up over time and become cumulative disadvantages (*Valian, 1999*; *Cole and Singer, 1991*). We observe a decreased growth in publications for all but the most productive men, and especially early-career researchers. This has the potential to reinforce disparities in an already heavily skewed system, if not given special attention, especially with regard to women. The widening gender gap in publishing observed in this study should thus be taken seriously by universities and funding agencies and factored into policies that allocate resources and support, as well as those that determine advancement and compensation, in order to mitigate inequities resulting from the unequal impact of the pandemic and its associated disruptions. Such inequities are deeply troubling both because they demonstrate how morally arbitrary characteristics like gender affect the opportunity to succeed in science and because they hinder the inclusion of diverse perspectives necessary to optimally advance scientific inquiry itself.

## Materials and methods

Data on authors and their publications. Publication data were retrieved from the Web of Science (WoS) in-house implementation at CWTS, Leiden University. This version of the WoS has linked tables between authors, their publications and information on the probable gender of authors.

The CWTS WoS includes a high-quality disambiguated table of authors and links to their publications. This list is produced through an algorithmic identification of publication clusters, using author, publication, source and citation data (*Caron and Eck, 2014*; *D'Angelo and van Eck, 2020*). This algorithm greatly improves the likelihood of an author profile containing the correct links to a scientist's publications, without including those of another author with the same name, and also including their own publications published under variations of their name. This algorithm so far has the highest precision and recall for this task (*Tekles and Bornmann, 2020*).

Author gender was inferred using a combination of Gender-API (https://gender-api.com/) and genderize (https://genderize.io/), in order to find the most likely gender of an author using their first name and country. The inferred gender is only applied in cases with >90% confidence, meaning gender ambiguous names, or names with very few observations for a country, are not

included. This leads to an exclusion of 20% of all authors, with a majority of those from China and South Korea, as first names in these countries tend to be less gendered than for most other countries.

Disciplines were inferred from the journal in which articles were published, using the translation table (http://help.prod-incites.com/inCites2Live/filterValuesGroup/researchAre-aSchema/oecdCategoryScheme.html) between WoS Subject Categories and the OECD Fields of Science from the Frascati Manual (OECD Working *OECD Working Party of National Experts on Science and Technology Indicators, 2007*). For each author, we summed the weighted major scientific fields and assigned the most frequent as their main discipline.

We queried the WoS for all authors with their first publication in either 2009 or 2010 (mid-career researchers) or 2016 or 2017 (early-career researchers). We excluded authors with fewer than three publications in total, and further limited the sample to authors with at least one publication in 2018 or 2019. The last step was done to create a sample of actively publishing scientists. We assigned main discipline codes to all authors and limited the sample to authors from *1.4 Chemical sciences*, *1.6 Biological sciences*, *3.1 Basic medicine* and *3.2 Clinical medicine*. This sample consisted of 431,207 authors linked to 2,113,108 publications in the period 2016–2020. The counterfactual sample was constructed identically, but for authors with their first publication in 2011 or 2012, counting their publications until 2015. This sample included 276,793 authors linked to 1,060,330 publications.

### Difference-in-differences model

To estimate the differential impact of the COVID-19 pandemic on the gender gap in publication productivity, we leveraged a difference-in-differences strategy. Because of a persistent gender gap in the number of publications over time, we used the yearly data on journal article publications prior to 2020 as baselines for estimating how the pandemic impacted the scholarly productivity of men and women differently. Although, not a randomized treatment, we treated the yearly gender difference in publication numbers (for 2016, 2017, 2018, and 2020) relative to the difference in 2019 as our key estimand. To estimate the average treatment effect on the treated ($ATT$), the gender difference relative to the baseline 2019 difference, we specified the following regression model:

$$Y_{it} = \alpha_i + \gamma_t + \sum_{t=-4}^{4} \delta_t \text{Gender}_i \cdot \text{Year}_t + \epsilon_{it} \qquad (1)$$

Where $Y_{it}$ denotes the number of published articles by individual in year $t$, $\alpha_i$ are the author fixed effects, $\gamma_t$ are the year fixed effects, and $\delta_t$ are a set of parameters with $t \in \{-4, -3, -2, 0\}$ estimating the difference in publication numbers between men and women each year, relative to the difference in 2019 ($t = -1$), which we left out of the estimation. The indicator $t$ is here the year relative to 2020. The $ATT$ for a given year $k$ relative to 2019 is then:

$$\begin{aligned} ATT_{t=k} = {} & E[Y^1_{\text{women}}|t=k] - E[Y^0_{\text{women}}|t=k] \\ & + \left[ E[Y^0_{\text{women}}|t=k] - E[Y^0_{\text{women}}|t=-1] \right] \\ & - \left[ E[Y^0_{\text{men}}|t=k] - E[Y^0_{\text{men}}|t=-1] \right] \end{aligned}$$
$$(2)$$

When used in the analysis, predicted values are the average partial effects at specified combinations of gender and year. We calculate the linear predicted value based on the regression model for each unit of observation (person i at year t), and average over these units for each specified subset of units (e.g. women in 2019 or men in 2018). This provides average predicted publications counts for each group at each time. Estimated differences in publication counts are the average marginal effects for each year derived from the regression model. The marginal effects are the partial derivative with respect to gender for each unit of observation, and the estimated average differences are then the mean of the unit-specific derivatives at each year.

### Parallel trends and counterfactual samples

Valid identification of the differential impact of the COVID-19 pandemic on researchers of different genders relies on a strong assumption of parallel trends of publication outcomes in pre-pandemic years. I.e. identification of the average treatment effect on women essentially assumes that $\left[ E[Y^0_{\text{women}}|t=k] - E[Y^0_{\text{women}}|t=-1] \right]$ $- \left[ E[Y^0_{\text{men}}|t=k] - E[Y^0_{\text{men}}|t=-1] \right] = 0$. A large literature (e.g. **Hart and Perlis, 2019**; **Mairesse and Pezzoni, 2015**) has documented persistent gender gaps in publication productivity. Our dynamic difference-in-differences model confirms this. A consistent gap between men and women is present in all years prior to 2020 for our full sample (**Figure 2**). This gap also tends to slightly increase over time, casting doubt on the assumption of similar publication trends for men and women scientists. **Figure 2—source data 1** shows a statistically significant difference in the publication gender gap between 2016 and 2019, and 2017 and 2019. However, the difference is much smaller, and statistically non-significant, when comparing 2018 and 2019.

We also modeled the differential publications rates for a counterfactual sample of researchers, who started publishing (or who's first publication was registered in the Web of Science database) in 2011, across the following five years. As shown in **Figure 3—figure supplement 1**, the gender gap in publication rates increased from almost parity in the first year to an average difference of 0.2 full publications five years after (0.05 fractionalized). Again, the gender gap increased with 1/20 of a full publication (full count: –0.05, 99% CI: [–0.0665; –0.0337], fractionalized count: –0.006, 99% CI: [–0.0094; –0.0028]) between four and five years after first publication, amounting to 53% of our $ATT$ from the full sample.

### Data on lockdown severity

To assess how the pandemic may entail different gender effects across countries and lockdown severity, we use data from the Oxford COVID-19 Government Response Tracker. We construct seven lockdown indicators at the country level by aggregating four measures of daily government COVID-policies across a whole year (from March 1st 2020 to December 31st 2020) in two ways. **Table 1** summarizes the seven indicators. We use four of the Oxford COVID-19 Government Response Tracker indicators (**Hale et al., 2021**) related to the coordinated close-downs of schools (C1) or workplaces (C2), stay at home requirements (C6), and the combined policy stringency index. First, we sum the indicator value across the whole year to create a cumulative sum of restriction severity for all four indicators, such that a lockdown indicator $L_k$ is the summarized values across 305 days:

$$L_k = \sum_{i=1}^{305} I_i \qquad (3)$$

Second, we count the number of days across the same period with the maximum indicator value for three indicators relating to school lockdowns, workplace lockdowns, and stay at home requirements. Each of these indicators can take the values 0, 1, 2, and three per day (where three

indicates the most severe policy situation for the three indicators in question). For these three indicators we create a conditional sum across 305 days. We then let $L_k$ be the number of days an indicator $I_1, ..., I_{305}$ equals 3:

$$L_k = \sum_{i=1}^{305} [I_i = 3] \tag{4}$$

Together, this gives us seven different indicators of lockdown severity at the national level. It is important to note that we use national-level policy indicators capturing only COVID-19 policy responses enacted at the country or federal level. In cases where sub-national policies supersede country-level restrictions, more or less severe policies are not reflected in the indicators.

### Heterogeneity in COVID-19 effects

To show the heterogeneity in possible COVID-19 induced treatment effects, we estimated our difference-in-differences model separately for each country, focusing on the 40 countries contributing 95% of all authors in our sample. We also investigated the degree to which this heterogeneity could be attributed to variations in the severity of policy restrictions across countries. Using the seven lockdown indicators described above, we compared country-level gender gaps with the measures of severity as shown in *Figure 5—figure supplement 1* and *Figure 5— figure supplement 2*.

**Emil Bargmann Madsen** is in the Danish Centre for Studies in Research and Research Policy, Aarhus University, Aarhus, Denmark
http://orcid.org/0000-0003-4394-5373
**Mathias Wullum Nielsen** is in the Department of Sociology, Copenhagen University, Copenhagen, Denmark
http://orcid.org/0000-0001-8759-7150
**Josefine Bjørnholm** is in the Danish Centre for Studies in Research and Research Policy, Aarhus University, Aarhus, Denmark
**Reshma Jagsi** is in the Department of Radiation Oncology, University of Michigan, Ann Arbor, United States
http://orcid.org/0000-0001-6562-1228
**Jens Peter Andersen** is in the Danish Centre for Studies in Research and Research Policy, Aarhus University, Aarhus, Denmark
jpa@ps.au.dk
http://orcid.org/0000-0003-2444-6210

*Author contributions:* Emil Bargmann Madsen, Conceptualization, Data curation, Formal analysis, Investigation, Methodology, Software, Validation, Visualization, Writing – original draft, Writing – review and editing; Mathias Wullum Nielsen, Conceptualization, Formal analysis, Funding acquisition, Investigation, Methodology, Project administration, Resources, Supervision, Validation, Writing – original draft, Writing – review and editing; Josefine Bjørnholm, Conceptualization, Data curation, Formal analysis, Investigation, Methodology, Software, Validation, Writing – review and editing; Reshma Jagsi, Conceptualization, Investigation, Supervision, Validation, Writing – original draft, Writing – review and editing; Jens Peter Andersen, Conceptualization, Data curation, Formal analysis, Funding acquisition, Investigation, Methodology, Project administration, Resources, Software, Supervision, Validation, Visualization, Writing – original draft, Writing – review and editing

*Competing interests:* The authors declare that no competing interests exist.

### Funding

| Funder | Grant reference number | Author |
|---|---|---|
| Samfund og Erhverv, Det Frie Forskningsråd | DFF-0133-00165B | Emil Bargmann Madsen Mathias Wullum Nielsen Josefine Bjørnholm Jens Peter Andersen |
| Aarhus Universitets Forskningsfond | AUFF-F-2018-7-5 | Jens Peter Andersen |
| Independent Research Fund Denmark | 9130-00029B | Mathias Wullum Nielsen |

The funders had no role in study design, data collection and interpretation, or the decision to submit the work for publication.

**Decision letter and Author response**
Decision letter https://doi.org/10.7554/eLife.76559.sa1
Author response https://doi.org/10.7554/eLife.76559.sa2

## Additional files

### Supplementary files
• Transparent reporting form

### Data availability
The current manuscript is a computational study, so no data have been generated for this manuscript. Source code and compiled data are available here: https://github.com/emilbargmann/covid_update (copy

archived at swh:1:rev:436c899ca98e80b5f09500bf54b-40b3649cc5b02) Raw data are available here: https://github.com/ipoga/covid19_gender.

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
