## [Decision Letter]

**Decision letter after peer review:**

Thank you for submitting your article "Individual-level researcher data confirm the widening gender gap in publishing rates during COVID-19" to *eLife* for consideration as a Feature Article. Your article has been reviewed by two peer reviewers, and the evaluation has been overseen by the *eLife* Features Editor, Peter Rodgers. The reviewers opted to remain anonymous.

The reviewers and editors have discussed the reviews and we have drafted this decision letter to help you prepare a revised submission. Please also note the following

i) If your article is accepted, we would like to publish some of the relevant code with the article: please let me know if this will be a problem.

ii) The *eLife* Features Editor will also contact you separately about some editorial issues that you will need to address regarding the appendixes and figure supplements.

Summary:

This very interesting paper estimates the differential impact of COVID-19 on the annual publication rates of male and female authors working in the disciplines of biology, chemistry, and clinical and basic medicine relative to the pre-pandemic publication rate. When the pandemic began early in 2020, a number of studies of preprints predicted that gender gaps in productivity would get larger. Using data on a global sample of 431,207 authors (2,113,108 publications), this paper documents a small but consistent average worsening in the gap between women's and men's annual publishing rates. The largest differential impacts by gender were observed for particularly prolific researchers, in the top-10% of the productivity distribution. The paper also performs heterogeneity analyses by career stage and discipline. In addition, the authors compared their results to a counterfactual sample of authors and their publications from 2011 to 2015, to see how much the gender gap widened over a 5-year period with no pandemic. Overall, this is a well-designed study with compelling results. However, there are a number of points that need to be addressed to make the article suitable for publication.

Major/Essential revisions:

1) I am curious as to why the authors decided to focus on the particular disciplines they do. In terms of generalizability of findings, this particular selection might present difficulties if these fields had different post-covid policy responses, as well as unique challenges due to the pandemic, as compared to others. It would be important to understand why these fields were selected and how this might affect the overarching conclusions.

2) The study looks at the annual publication rate pre and post covid, focusing on how the gender gap in 2020 compared to 2019. The disruptions due to the pandemic did not start in most parts of the world until March of 2020. I would encourage the authors to conduct the analysis at a more granular level (by, say, looking at monthly publications rather than annual averages)? This could also potentially enrich the country-level analysis, as there was considerable time variable in lockdowns.

Note from editor: Addressing this point is optional, but I would encourage you to perform this analysis if possible. If you decide not to address this point, please discuss it as a topic that could be explored in future research.

3) One of the contributions that the paper emphasizes on p. 1 line 30 is that the study considers longer-term publication impacts. I would caution against using "long-term" to refer to 2020. As the paper rightly points out on p. 9, line 212, publishing takes quite a long time. From the start of the research project to the publication stage, we might expect to see much longer lags than just a few months. Even in the sciences, where the publication process is quicker than in other disciplines, it's highly unlikely that we are seeing an impact on new projects. The 2020 papers being published are likely old projects that were already in the publication pipeline and are now coming out at journals. Therefore, these are much more likely to pick up the pre-pandemic trend, which we see in fact – Figure 2 shows a slight widening of the gender gap pre-treatment (from -.2 to roughly around -.27 or so). I would encourage the authors to reframe the article with this in mind.

4) The analysis can be expanded to look at the quality dimension of publications (control for journal rankings in some systematic manner), as well as how the female authors stack up against male authors in terms of contributions. In science, we can tell clearly which author contributed the most to the project (first author). It would be interesting to see an analysis by first author only.

Note from editor: Addressing this point is optional: however, if you decide not to address it please discuss it as a topic that could be explored in future research.

5) I would like to know more details about the country-level comparisons. In a recent NBER Working Paper 29668, Deryugina et al. (2022) show that school openness at the country level correlates with a reduction in the gender gap in research time lost by men and women. It would also be important to know how the data sources treats partial closures, timing of closures, as well as academic breaks.

6) As regards severity, I am not sure why the authors lump countries into severity bins rather than using country-level openness measures. There also needs to be greater discussion of the results of the lockdown severity analysis. Why did you find no pattern? Is it because there is too much within-country heterogeneity in the timing, type, and severity of lockdown measures to observe a pattern, and finer-grain analyses (e.g., on a city-by-city basis) are needed? Is it that the demographics of women scientists differ substantially among countries, affecting, for example, the fraction of women scientists with young children? Is it that institutional responses and supports counteracted lockdown effects in some places but not others?

7) My main criticism is that the paper's Discussion section is rather boilerplate; it covers much of the same ground as previous papers on the pandemic's effects on women academics. Instead, I think I would be better to highlight some of the unique insights derived from the present analysis that currently get little attention in the Discussion. For example, I was struck by the analysis in Figure 4, which clearly shows that the gender gap is widest among the most productive scientists. This was true before COVID-19 and has become more pronounced during the pandemic. Why is this? What does it suggest about the pandemic's impacts on women versus men scientists? At the risk of over-interpreting Figure 4, it appears from panel B that the most productive men either maintained or improved their productivity during the pandemic, while the pandemic hurt the productivity of everyone else. Are highly productive men, but not women, most likely to have partners that do not work and who could thus shoulder additional caregiving or domestic duties? Are highly productive men, but not women, most likely to be spared the additional teaching or service burdens during the pandemic? I realize the present analysis cannot answer these questions, but the authors should discuss these possibilities.

8) Several figures (Figure 2 and the early-career panel of Figure 3, as well as the fractional-count versions in the supplement) suggest that publication rate declined for both men and women authors during the pandemic, it just declined more among women. This is worth discussing as it suggests that tenure committees and grant panels, etc., need to revise their expectations vis-à-vis productivity for all researchers, and especially so for women scientists.

9) The discussion says, "unlike prior studies, we find that the gendered effects of COVID-19 are salient for early career-scientists with four years of publication experience as well as for mid-career scientists with ten years of publication experience" (Lines 154-156), but I found this a bit misleading. Similarly, Lines 16-17 in the abstract imply that the gender gap widened similarly for early-career and mid-career scientists. But from Figure 3, it appears there is a growing gender gap during the pandemic for both early-career and mid-career researchers, but that the gender gap grows larger for mid-career than for early-career scientists. Please rephrase the abstract and discussion to reflect Figure 3.

---

## [Author Response]

Major/Essential revisions:1) I am curious as to why the authors decided to focus on the particular disciplines they do. In terms of generalizability of findings, this particular selection might present difficulties if these fields had different post-covid policy responses, as well as unique challenges due to the pandemic, as compared to others. It would be important to understand why these fields were selected and how this might affect the overarching conclusions.

Thank you for this comment. We agree, there needs to be a justification of the field selection, which we added in the introduction, as follows:

“We chose these fields as they are well-represented in Web of Science (more than 90% of references are included in Web of Science), their primary knowledge production mode is through journal publication (as opposed to e.g. computer science and most engineering fields, or the humanities), research is comparatively collaborative (although some types of clinical research has somewhat more authors), publishing is relatively fast (compared to the social sciences). Biology, clinical and basic medicine also have some of the highest shares of women scientists in the natural sciences.”

2) The study looks at the annual publication rate pre and post covid, focusing on how the gender gap in 2020 compared to 2019. The disruptions due to the pandemic did not start in most parts of the world until March of 2020. I would encourage the authors to conduct the analysis at a more granular level (by, say, looking at monthly publications rather than annual averages)? This could also potentially enrich the country-level analysis, as there was considerable time variable in lockdowns.Note from editor: Addressing this point is optional, but I would encourage you to perform this analysis if possible. If you decide not to address this point, please discuss it as a topic that could be explored in future research.

During the planning of this study, we intended to do precisely this. However, data on publication month is surprisingly not available or representable for all publications (and even when it is, journals may use labels such as “Summer 2020”). In addition, while we look into the effect of lock-downs (and don’t find a clear one), there are many other reasons the pandemic has influenced the publication productivity of men and women differently, e.g. differences in funding, teaching workloads, burnout, expectations of academic service and care etc.

As a final reason behind the choice not to use per-month data, is the fact that for most scientists, publishing a paper is a somewhat rare event that does not happen on a monthly basis. We would not have been able to use the panel-based difference-in-difference design allowing this type of analysis, if we should have analysed differences on a per-month basis.

We have added the following as a limitation in the discussion:

“Our analysis focused on annual publishing rates, which may obscure some of the potential effects of e.g. school closures on the immediate publishing rates. A more granular analysis of monthly publishing rates may reveal a more direct correlation between lock-downs and decreased publishing rates. However; information on when something is published is not available on a monthly basis for a large proportion of articles, and information on submission and review dates are even harder to obtain, often completely missing even through manual process. Furthermore, many of the delays in the publishing process are out of the hand of authors and thus unrelated to the lock-down effect they may be experiencing, and many other pandemic factors may be related to an increasing gender gap than the very specific lock-downs. By looking at annual data, we see a more robust effect overall. We strongly encourage publishers to publish transparent, open machine- and human-accessible information on which day a manuscript was received, reviewed, revised, accepted and published.”

3) One of the contributions that the paper emphasizes on p. 1 line 30 is that the study considers longer-term publication impacts. I would caution against using "long-term" to refer to 2020. As the paper rightly points out on p. 9, line 212, publishing takes quite a long time. From the start of the research project to the publication stage, we might expect to see much longer lags than just a few months. Even in the sciences, where the publication process is quicker than in other disciplines, it's highly unlikely that we are seeing an impact on new projects. The 2020 papers being published are likely old projects that were already in the publication pipeline and are now coming out at journals. Therefore, these are much more likely to pick up the pre-pandemic trend, which we see in fact – Figure 2 shows a slight widening of the gender gap pre-treatment (from -.2 to roughly around -.27 or so). I would encourage the authors to reframe the article with this in mind.

This is a good point. We have changed longer-term effects to longitudinal effects in accordance with your suggestion.

4) The analysis can be expanded to look at the quality dimension of publications (control for journal rankings in some systematic manner), as well as how the female authors stack up against male authors in terms of contributions. In science, we can tell clearly which author contributed the most to the project (first author). It would be interesting to see an analysis by first author only.Note from editor: Addressing this point is optional: however, if you decide not to address it please discuss it as a topic that could be explored in future research.

Thank you for the suggestion. We added a descriptive analysis of the trends in women’s first authorship shares between 2016 and 2020 for each of the four disciplines. For each year, women were more likely to occupy the first authorship compared to a baseline of all authorships, showing no clear decline in women’s first authorships. Only chemistry showed a slight decline in women’s first authorships relative to all authorships, but these are not dramatic changes. We have added this clarification to the Discussion section.

5) I would like to know more details about the country-level comparisons. In a recent NBER Working Paper 29668, Deryugina et al. (2022) show that school openness at the country level correlates with a reduction in the gender gap in research time lost by men and women. It would also be important to know how the data sources treats partial closures, timing of closures, as well as academic breaks.

Thank you for the suggestion and relevant citation. We have added considerations in the Discussion section specifically focusing on why lock-down severity does not seem to correlate to gender gap levels. A likely factor is the need to aggregate day-to-day levels of policy severity across almost the whole year of 2020 (305 days, as stipulated in the Materials and methods section) to compare these to yearly publication gaps. Furthermore, we have clarified that these indicators pertain to the national policy level, but ignoring sub-national variation. This may be inappropriate in some countries where Covid-19 policies have largely been enacted at state or regional levels. However, with such a large country sample it would be impossible to compare gender gaps across countries where relevant policies are mostly sub-national or nation. As our sample comprises many researchers from the US, an idea for future studies could be to conduct a case study on individual researchers in US universities and state-level genders gaps vis-à-vis Covid-policies. This is, nevertheless, beyond the scope of this article.

6) As regards severity, I am not sure why the authors lump countries into severity bins rather than using country-level openness measures. There also needs to be greater discussion of the results of the lockdown severity analysis. Why did you find no pattern? Is it because there is too much within-country heterogeneity in the timing, type, and severity of lockdown measures to observe a pattern, and finer-grain analyses (e.g., on a city-by-city basis) are needed? Is it that the demographics of women scientists differ substantially among countries, affecting, for example, the fraction of women scientists with young children? Is it that institutional responses and supports counteracted lockdown effects in some places but not others?

Thank you for this comment. We have changed the severity-related figure to instead show the individual country-level gender gaps plotted against the openness measures. The revised figure show a similar lack of relationship to national gender gaps in publications. As stated above, we think it highly likely that within-country, and across-time, variation in Covid-response policies makes it hard to discern any decisive patterns from the more aggregate data. A city-by-city analysis would not be feasible as large scale data on city policies are not available, and given the lack of wide-covering data on monthly publications, a more fine-grained analysis over time is not possible either.

7) My main criticism is that the paper's Discussion section is rather boilerplate; it covers much of the same ground as previous papers on the pandemic's effects on women academics. Instead, I think I would be better to highlight some of the unique insights derived from the present analysis that currently get little attention in the Discussion. For example, I was struck by the analysis in Figure 4, which clearly shows that the gender gap is widest among the most productive scientists. This was true before COVID-19 and has become more pronounced during the pandemic. Why is this? What does it suggest about the pandemic's impacts on women versus men scientists? At the risk of over-interpreting Figure 4, it appears from panel B that the most productive men either maintained or improved their productivity during the pandemic, while the pandemic hurt the productivity of everyone else. Are highly productive men, but not women, most likely to have partners that do not work and who could thus shoulder additional caregiving or domestic duties? Are highly productive men, but not women, most likely to be spared the additional teaching or service burdens during the pandemic? I realize the present analysis cannot answer these questions, but the authors should discuss these possibilities.

This is a fair point, and we have made a number of revisions based on the other comments which we believe alleviates some of the “boilerplate” phrasing. To address the prolific-perspective, we also added the following paragraph to the discussion:

“Those designing interventions to promote equity in academic science and medicine should strive to understand the reasons why highly prolific men appeared able to maintain their annual publication rates while highly prolific women were not. Prior research suggests that it is possible that men with the highest levels of productivity may have been more likely to have been rewarded with access to additional workplace supports, such as endowed professorships, in recognition of their achievements (Gold et al. 2020). If so, this might have served as a cushion against the impact of the pandemic on those individuals. Moreover, if institutions prioritized protecting a few “superstar” researchers from teaching or clinical demands without clear processes for identifying which individuals received preferential treatment, the vast literature on unconscious bias suggests that such efforts might preferentially have protected outstanding men as compared to similarly outstanding women (NASEM, 2007). Prior research also suggests that high-achieving women scientists may be more likely than their male peers to state that their partners’ careers take priority (Mody et al., 2020). Indeed, it is possible that high-achieving men scientists’ partners may be particularly likely to be willing to make sacrifices in their own careers to take on additional domestic labor to allow continuation of their extraordinary partners’ work. If partners of extraordinarily productive women scientists are less willing to do so, and if this difference is even more marked than any differences that may exist when a scientist is less highly productive, this could also serve as a mechanism to drive the differences observed. Further research is necessary to investigate these and other possibilities.”

8) Several figures (Figure 2 and the early-career panel of Figure 3, as well as the fractional-count versions in the supplement) suggest that publication rate declined for both men and women authors during the pandemic, it just declined more among women. This is worth discussing as it suggests that tenure committees and grant panels, etc., need to revise their expectations vis-à-vis productivity for all researchers, and especially so for women scientists.

Thank you for this comment. We agree with the observation, and that grant panels, promotion committees etc. need to be aware of this. However, in many of those cases, comparisons between candidates occur at the same career stage and when not, evaluators are aware of career stage differences. We therefore consider the increasing gender gap more crucial and still focus our discussion on this. We included the following in the concluding paragraph of the discussion to give attention to the more overall problem:

“We observe a decline in publications for all but the most productive men, and especially early career researchers. This has the potential to further skew an already heavily skewed system, if not given special attention, especially with regard to women.”

9) The discussion says, "unlike prior studies, we find that the gendered effects of COVID-19 are salient for early career-scientists with four years of publication experience as well as for mid-career scientists with ten years of publication experience" (Lines 154-156), but I found this a bit misleading. Similarly, Lines 16-17 in the abstract imply that the gender gap widened similarly for early-career and mid-career scientists. But from Figure 3, it appears there is a growing gender gap during the pandemic for both early-career and mid-career researchers, but that the gender gap grows larger for mid-career than for early-career scientists. Please rephrase the abstract and discussion to reflect Figure 3.

Thank you for pointing to this inconsistency in the representation and discussion of our results. While the numerical difference is indeed larger for mid-career scientists, the relative change in the gender gap is larger for early career scientists. We have revised the results and Discussion sections to make this point clearer.